# Straw-Enhanced Soil Bacterial Robustness via Resource-Driven Niche Dynamics in Tea Plantations, South Henan, China

**DOI:** 10.3390/microorganisms13040832

**Published:** 2025-04-06

**Authors:** Xiangchao Cui, Dongmeng Xu, Yu Zhang, Shuping Huang, Wei Wei, Ge Ma, Mengdi Li, Junhui Yan

**Affiliations:** 1Henan Key Laboratory for Synergistic Prevention of Water and Soil Environmental Pollution, School of Geographical Sciences, Xinyang Normal University, Xinyang 464000, China; dongmengxu1221@gmail.com (D.X.); huangshuping8808@126.com (S.H.); weiweiqyz@xynu.edu.cn (W.W.); mage@xynu.edu.cn (G.M.); lmd20210605@163.com (M.L.); yanjh2015@126.com (J.Y.); 2College of Plant Protection, Shandong Agricultural University, Tai’an 271001, China; yuzhang@sdau.edu.cn; 3North-South Transitional Zone Typical Vegetation Phenology Observation and Research Station of Henan Province, Xinyang Normal University, Xinyang 464000, China

**Keywords:** straw return, tea soil, bacterial community, biomarker, co-occurrence network

## Abstract

Straw application (SP) is a promising strategy for the improvement of soil fertility, but the biological effects and the mechanisms of its effects on microorganisms remain unclear. The investigation into the tea plantations (CK/S) in southern Henan, China, without/with straw amendment was carried out to assess the effects of SP on the soil bacterial communities using high-throughput sequencing. SP induced the community restructuring of the dominant phyla, e.g., Acidobacteriota, Pseudomonadota, Chloroflexota, with significantly increasing Nitrospirota, Vicinamibacterales and Anaerolineaceae (*p* < 0.05), while reducing Terriglobales (*p* < 0.05). These transitions correlated with significantly enhanced *α*-diversity and *β*-diversity divergence (*p* < 0.05). The linear discriminant analysis effect size (LEfSe) results confirmed the significant selective enrichment of nitrogen-cycling taxa (*Nitrospira*), copiotrophs (*Chryseotalea*), and anaerobic degraders (Anaerolineaceae), along with the suppression of the oligotrophic lineage (*Ellin6067*) by SP (*p* < 0.05). The co-occurrence networks of S had lower topological properties and negative cohesion (*p* < 0.05), which exhibited intensified simplified complexity and competition. The soil water content (WC) and pH were the main drivers of *β*-diversity variation and the keystone taxa assembly, as calculated out by distance-based redundancy analysis (dbRDA). This study demonstrates that SP can enhance bacterial network stability and functional redundancy by resource-driven niche partitioning between copiotrophic taxa and nitrogen-cycling guilds through a competition–cooperation equilibrium.

## 1. Introduction

Straw return, a pivotal approach for agricultural waste recycling, has demonstrated significant effects in ameliorating the physicochemical properties of soils, regulating microbial community structure and enhancing soil ecosystem functions [1]. Accumulating evidence confirms that straw application (SP) alters critical soil parameters including carbon-to-nitrogen (C/N) ratios and nutrient availability [1,2,3,4]. Although straw application introduces exogenous microorganisms, studies have shown that its primary influence on soil microbial communities is mediated through alterations to the soil’s physicochemical properties rather than the direct effects of these external microbes on the resilience of soil microbial communities [5]. Rich in organic matter and essential nutrients (N, P, K), straw decomposition profoundly influences microbial metabolic pathways by modifying soil C/N stoichiometry [2,3]. Recent investigations have revealed that SP enhances soil microbial *α*-diversity indices while inducing structural shifts through the selective enrichment of copiotrophic populations (e.g., Proteobacteria) and the suppression of oligotrophic groups (e.g., Acidobacteria) [4]. As a consequence of the perennial monoculture practices and acidic soil conditions of the tea plant (*Camellia sinensis*), long-term monocropping in tea plantations has resulted in reduced soil disturbance, while the acidic soil conditions impose strong selective pressures on soil microbiota [6,7]. Thus, the tea plantation soils are characterized by unique microbial dynamics [8,9]. Although the functional mechanisms of SP in annual crop ecosystems (e.g., *Triticum aestivum* or *Zea mays*) have been well characterized [4], the ecological effects on tea plantations remain elusive.

Soil microorganisms, functioning as ecosystem engineers, maintain soil multifunctionality through organic matter decomposition and nutrient cycling regulation [10,11,12,13]. Emerging evidence in annual crop ecosystems indicates that the stability of a microbial community exhibits positive correlations with diversity indices, species asynchrony, and metabolic functional redundancy [13,14,15]. Notably, exogenous organic inputs may disrupt community stability by altering the topological properties of keystone microbiota networks, particularly through reduced modularity and shifted node centrality distributions [10]. It has been found that organic fertilizer use in Yunnan tea plantations enhances microbial diversity and ecological network stability [16]. The application efficacy of straw and its organic application may exhibit spatiotemporal variability due to soil heterogeneity and significant climatic specificity-induced variations in the microbial communities of tea plantation soils [17]. The ecological effects of SP on the microbial ecosystem remain unclear in those tea plantations with different soil types and climates; e.g., the tea plantations in southern Henan, China, characterized by acidic yellow–brown soils, high precipitation, and distinct seasonal temperature fluctuations.

Through field experiments conducted in the core production region of Xinyang Maojian (one of China’s Top Ten Famous Teas) [18], this study investigated three mechanistic questions: (1) How does SP differentially alter microbial community structure between rhizospheres and bulk soils, particularly regarding *α*-diversity metrics (Shannon et al.), *β*-diversity patterns (Bray–Curtis dissimilarity), and biomarker composition? (2) What network-level impacts does SP exert, including changes in topological properties (node degree distribution and modularity index) and keystone taxa interactions? (3) Which environmental factors (pH, organic carbon, and nitrogen/phosphorus availability) mediate straw-induced shifts in microbial assembly processes and network resilience? By integrating high-throughput sequencing, molecular ecological network analysis, and multivariate statistics, this work can provide actionable advice for sustainable tea garden management by advancing our understanding of soil microbiome engineering in perennial agroecosystems.

## 2. Materials and Methods

### 2.1. Description of Sites and Sampling

The experimental area is located in the town of Dongjiahe (32°02′21″–32°23′23″ N, 113°45′28″–114°13′09″ E), Shihe District, Xinyang City, Henan Province, China, a primary cultivation zone for *Camellia sinensis* (Xinyang Maojian tea) (Figure 1). This region lies within the transitional ecotone between the northern subtropical and warm temperate monsoon climate zones, characterized by distinct seasonal variations in temperature and precipitation [19]. The 2018 meteorological records indicate an annual mean temperature of 16.6 °C, with summer and winter averages of 27.8 °C and 3.4 °C, respectively. The mean annual precipitation is 992 mm, and the relative humidity averages 74.7% (Appendix A). The topography comprises eroded hills and low-mountain terrains (elevation: 100–800 m), with dominant yellow–brown soils classified as Haplic Luvisols (FAO taxonomy), with the soil texture classes being light loam and sandy loam, exhibiting a pH range of 4.7–6.5 [20]. Local tea plantations adhere to a standardized fertilization protocol—basal applications of calcium superphosphate (Ca(H_2_PO_4_)_2_, 375–750 kg·ha^−1^) and potassium sulfate (K_2_SO_4_, 225–375 kg·ha^−1^) in October–November, followed by nitrogen-focused topdressings of ammonium sulfate ((NH_4_)_2_SO_4_, 37–125 kg·ha^−1^) in mid-February and post the spring harvest.

Soil samples were collected in April 2018 from four independent tea plantations in Dongjiahe Town (Figure 1). The selection of the sampling time was based on the relatively stable precipitation during this period, which is comparable to over half of the months annually, which ensured that the soil water content reflects the soil’s water capacity to a certain extent [21]. Wheat straw mulching was applied between the tea plant rows in tea plantations 1 and 3, whereas rice straw mulching was used in tea plantations 2 and 4 (Appendix A). Both the wheat and rice straw lengths were controlled to 15–20 cm. Each plantation included two treatments: (1) a control (CK) with no SP for 10 years, and (2) the straw application (S) with annual straw mulching (15,000 kg·ha^−1^ fresh matter) for 10 years. A randomized stratified sampling strategy was employed to ensure spatial representativity. For each treatment, three replicate plots (10 m × 10 m) were established. After removing the 0–5 cm topsoil to eliminate interference from external environmental factors, five subsamples (5–20 cm depth) per plot were collected diagonally within each plot using a stainless-steel auger (with a 5 cm diameter). Subsamples from the same plot were homogenized and brought back to the laboratory for the subsequent microbial molecular sequencing and the determination of the soils’ physicochemical properties. A total of 24 composite samples (4 plantations × 2 treatments × 3 replicates) were obtained for further analysis.

### 2.2. Soil Physicochemical Property Analysis

The samples were air-dried and homogenized by sieving through a 2 mm mesh sieve to remove plant materials and gravel (Appendix A). The soil gravimetric water content (WC) was determined by mass loss after oven-drying at 105 °C until constant weight (±0.01 g) [22]. For the pH and electrical conductivity (EC) measurements, soil suspensions were prepared using pre-boiled deionized water (boiled ≥ 30 min to eliminate dissolved CO_3_^2−^) at soil/water ratios of 1:2.5 (*w*/*w*) and 1:5 (*w*/*w*), respectively, with measurements conducted after 30 min of equilibration [22]. The available phosphorus content (AP) was extracted using the Bray-I method (0.03 mol·L^−1^ NH_4_F + 0.025 mol·L^−1^ HCl), and quantified via molybdenum–antimony colorimetry [23]. Soil organic matter (SOM) content was determined by dichromate oxidation [24]. Soil NH_4_^+^-N and NO_3_^−^-N contents were extracted by 2 mol L^−1^ KCl and determined by a Continuous Flow Analytical System (Skalar San^++^) [25]. Cations (Ca^2^^+^, Al^3^^+^) were treated by alkali fusion, and analyzed via the inductively coupled plasma-atomic emission spectrometer (ICP-AES) [26].

### 2.3. DNA Extraction, PCR, and Illumina Sequencing

The genomic DNA was extracted from 0.5 g of soil sample using FastDNA SPIN Kit (MP Biomedicals, Santa Ana, CA, USA), with the purity and concentration verified spectrophotometrically (NanoDrop ND-1000, Thermo Scientific, Wilmington, DE, USA). The V4–V5 hypervariable regions of the bacterial 16S rRNA genes were amplified using barcoded primers 515F (5′-GTGCCAGCMGCCGCGGTAA-3′) and 909R (5′-CCCCGYCAATTCMTTTRAGT-3′) in 50 μL PCR reactions [27]. The thermocycling conditions included initial denaturation at 95 °C for 3 min, 30 cycles of 95 °C for 30 s, 55 °C for 30 s, and 72 °C for 1 min, followed by a final extension at 72 °C for 6 min. Triplicate reactions per sample were pooled, purified (QIAquick PCR Purification Kit, QIAGEN, Venlo, Netherlands), and quantified fluorometrically using NanoDrop ND-1000 (Thermo Scientific, Wilmington, DE, USA). Then, the DNA samples were sequenced by means of an Illumina HiSeq X Ten System platform (Illumina, San Diego, CA, USA) following the manufacturer’s protocols.

### 2.4. Data Analysis

Microbial community analysis was performed using the 16S rRNA data generated by the quantitative insights into microbial ecology 2 (QIIME2). The artifacts (.qza) were converted into phyloseq objects using the qiime2R package (v0.99.20) (https://github.com/jbisanz/qiime2R.git, accessed on 24 March 2020) for downstream statistical analysis. Statistical analyses were performed in R 4.4.1. The community composition of the dominant bacteria was analyzed using Excel 2021 (Appendix A), based on the data of the amplicon sequence variants (ASVs) obtained from the phyloseq objects, and drawn by OriginPro 2024b. The significance of the community composition of the dominant bacteria was analyzed using Mann–Whitney U tests. The *α*-diversity was estimated through Shannon, abundance-based coverage estimator (ACE), observed OTUs (OBS), and phylogenetic diversity (PD), which was computed with the picante package [28]. And, the *β*-diversity index was analyzed by principal coordinate analysis (PCoA) based on the Bray–Curtis distances using the vegan package [29]. The Shapiro–Wilks test was conducted to test the normality distributions of *α*-diversity. Based on the non-normality results of *α*-diversity, the significance of *α*-diversity and *β*-diversity was analyzed via Student’s *Wilcoxon* test, respectively, and the permutational multivariate analysis of variance (PERMANOVA) test.

The linear discriminant analysis effect size (LEfSe) analysis identified biomarkers between experimental groups (CK vs. S) using the microeco package, with significance determined by Kruskal–Wallis and Wilcoxon tests [30]. Co-occurrence networks were reconstructed using SpiecEasi [31] and visualized in Gephi (0.10.1), with correlation matrices generated using the Hmisc package [32]. The keystone taxa were identified via ggClusterNet, with the *Z_i_*-*P_i_* threshold criteria—module hubs (*Z_i_* > 2.5, *P_i_* < 0.62), connectors (*Z_i_* < 2.5, *P_i_* > 0.62), and network hubs (*Z_i_* > 2.5, *P_i_* > 0.62) [33,34]. Community cohesion was quantified using ASV correlation matrices and the relative abundance of ASVs [32]. Environmental drivers were confirmed through distance-based redundancy analysis (dbRDA) [35] and variance partitioning with the rdacca.hp package [36]. Permutation tests were applied to validate statistical significance.

## 3. Results

### 3.1. Community Composition of Dominant Bacteria

A total of 100,690 high-quality sequences were clustered into 22,668 ASVs. The dominant bacterial taxa (relative abundance ≥ 1%) were taxonomically resolved as shown in Figure 2. At the phylum level (Figure 2a), Acidobacteriota, Pseudomonadota, and Chloroflexota constituted the core microbiota across treatments. S specifically elevated Nitrospirota’s relative abundance compared to the control (CK) (*p* < 0.05), while other phyla remained statistically invariant (*p* > 0.05).

At the family level (Figure 2b), the dominant bacterial families mainly consist of Nitrosomonadaceae, Incertae_Sedis_o_Terriglobales, Gemmatimonadaceae, Incertae_Sedis_o_Vicinamibacterales, Incertae_Sedis_o_Subgroup_2, Anaerolineaceae, Xanthobacteraceae, Incertae_Sedis_o_Gaiellales, Pyrinomonadaceae, and Vicinamibacteraceae. Notably, S significantly enriched Incertae_Sedis_o_Vicinamibacterales (*p* < 0.05) and Anaerolineaceae (*p* < 0.05), concomitant with a 67% reduction in Incertae_Sedis_o_Terriglobales (*p* < 0.05). These shifts demonstrate SP’s capacity to restructure dominant bacterial assemblages across taxonomic hierarchies.

### 3.2. α- and β-Diversity of Bacterial Communities

The *α*-diversity analysis revealed the significant effects of SP (Figure 3a). S increased the Shannon, ACE, OBS, and PD indices by 3.7%, 12.0%, 12.1%, and 11.2%, respectively. Compared to CK, S significantly increased the Shannon index (*p* < 0.05). Notably, ACE, OBS richness, and phylogenetic diversity (PD) exhibited the most pronounced enhancement under S treatment (*p* < 0.01).

The *β*-diversity analysis, based on Bray–Curtis distances at the ASV level (Figure 3b), showed the shifts in the bacterial community distribution between S and CK. The first two principal coordinates (PCoA1: 22.3% and PCoA2: 13.7%) explained 36.0% of the total variance. The S samples demonstrated tighter clustering, corroborated by significant PERMANOVA differentiation (*F* = 1.76, *p* = 0.043). These results collectively demonstrate that S enhanced soil bacterial *α*-diversity and altered the *β*-diversity structure.

### 3.3. Biomarker Analysis

The LEfSe analysis revealed 56 differentially abundant bacterial taxa (LDA > 3.0, *p* < 0.05), with 40 S-enriched and 16 CK-enriched taxa (Appendix A). The top 30 discriminative taxa spanned 8 phyla, where Acidobacteriae (LDA = −4.47) in CK and Anaerolineae (LDA = 4.11) in S emerged as the primary biomarkers (Figure 4a). The phylum-level analysis confirmed Nitrospirota as the differential bacterial taxa (*p* < 0.05). The genus-level analysis demonstrated treatment-specific patterns—the uncultured genus *Ellin6067* (LDA = −3.63) dominated CK soils (*p* < 0.05), while *Chryseotalea* (LDA = 3.60), *Pirellula* (LDA = 3.53), *Nitrospira* (LDA = 3.41), and *MND1* (LDA = 3.82) were significantly enriched in S (*p* < 0.05).

The cladogram analysis of the top 30 taxa (Figure 4b) mapped differential abundance patterns across eight phyla (Acidobacteriota, Bacteroidota, Candidatus Eremiobacterota, Chloroflexota, Myxococcota, Nitrospirota, Planctomycetota, and Pseudomonadota), and six genera (*Chryseotalea*, *Nitrospira*, *Incertae_Sedis*, *Pirellula*, *MND1*, *Ellin6067*). Notably, SP suppressed *Ellin6067*, while stimulating *Nitrospirota* and copiotrophic genera like *Chryseotalea* and *Pirellula*.

### 3.4. Soil Bacterial Co-Occurrence Network Analysis

The microbial co-occurrence networks were constructed based on Spearman’s rank correlation analysis (|*R*| ≥ 0.8, *p* < 0.01) after false discovery rate (FDR) adjustment using dominant bacterial genera (prevalence >20% and total relative abundance >0.01% across samples). Both the network analysis of CK and S revealed a highly modular architecture, characterized by a modularity index (*Q_CK_* = 0.663, *Q_S_* = 0.687) with a high clustering coefficient (*C_CK_* = 0.573, *C_S_* = 0.530) (Figure 5a and Table 1). S reduced network complexity through decreased edges (36.8%), average degree (35.1%), graph density (29.4%), and network diameter (9.52%), with the negative edges between nodes increased by 69.2%. Positive edges dominated both networks (CK: 98.7%; S: 97.8%), indicating the prevalent cooperative interactions of the bacteria in both CK and S. Modularity analysis partitioned the CK and S networks into seven and five modules, respectively, with Acidobacteriota, Pseudomonadota, and Chloroflexota serving as the primary phylogenetic drivers between modules (Table 2).

The keystone taxa of CK and S are displayed in Figure 5b. The CK network exhibited 17 module hubs and 1 connector distributed among Acidobacteriota (33.3%), Chloroflexota (22.2%), Actinomycetota (16.7%), and Myxococcota (11.1%). In contrast, S induced a marked shift in keystone composition, with the S network containing four module hubs and five connectors, predominantly concentrated in Pseudomonadota (44.4%), Acidobacteriota (33.3%), Myxococcota (11.1%), and Gemmatimonadota (11.1%). This restructuring enhanced among-module connectivity with the *P_i_* average of S increasing from 0.08 to 0.40, and reduced within-module connectivity with the *Z_i_* average of S decreasing from 3.30 to 1.96, which was achieved by Pseudomonadota, Chloroflexota, Actinomycetota, and Gemmatimonadota, with Acidobacteriota altered at the class and order levels (Appendix A).

The bacterial community cohesion was analyzed to reveal the nuanced patterns of the microbial network between CK and S. Though positive community cohesion (Figure 6a) showed no significant difference between CK (0.250) and S (0.264) (*p* > 0.05), the negative cohesion (Figure 6b) exhibited an extremely marked divergence by 13.9% (*p* < 0.001) (Figure 6). This suggests that S treatment selectively intensified competitive microbial interactions without significantly altering cooperative networks.

### 3.5. Soil Factors Influencing Microbial Communities

The dbRDA results related to the bacterial *β*-diversity (Figure 7a) reveal that soil physicochemical factors explain 58.4% of the variation. Hierarchical partitioning (HP) identified WC, pH, and AP as the primary drivers with the individual percentages of, respectively, 24.7%, 20.6%, and 11.5% (Table 3). And, the permutation test also confirmed the significant effects (*p* < 0.05) of pH and WC and AP (Appendix A). According to the results, WC was the key driving factor of the S variation, while pH was the main driver of the CK variation, with AP functioning both in CK and S.

For the dbRDA results related to the keystone taxa (Figure 7b), the soil factors contributed 68.6% of the variation. Both pH and WS had significant effects (*p* < 0.05) based on the permutation test (Appendix A). Thus, WC (30.7%) and pH (21.0%) were identified as major contributors to the keystone taxa (Table 3). Furthermore, the keystone taxa exhibited similar responses to soil pH and WC as those observed for the bacterial *β*-diversity, indicating the keystone taxa community as a robust indicator of bacterial community assembly.

## 4. Discussion

This study comprehensively analyzed the effects of SP on tea rhizosphere soil bacterial communities using high-throughput sequencing and bioinformatics. The soil microbiome demonstrates strong resilience, with straw-induced shifts primarily driven by the straw’s characteristics rather than exogenous microbial inputs, as the latter fail to persist under prolonged soil conditions [5]. The SP treatment significantly altered the composition of the soil microbial communities. At the phylum level, the relative abundance of Nitrospirota in straw-amended rhizosphere soil was significantly higher than that in CK (*p* < 0.05). As Nitrospirota is recognized for its involvement in nitrite oxidation and complete ammonia oxidation (Comammox) metabolism [37,38], the relative abundance increase in Nitrospirota suggests that SP facilitates nitrogen nutrient supply for tea plants by reducing the ammonia volatilization and increasing the nitrate nitrogen and maintaining favorable physicochemical conditions in the soil [39]. At the family level, S-treated soil showed significant increases in Vicinamibacteraceae and Anaerolineaceae (*p* < 0.05), by 63.7% and 81.4%, while o_Terriglobales_Incertae_edis exhibited a decreased abundance (*p* < 0.05) of 50.2%. Vicinamibacteria demonstrates phosphorus solubilization potential, Anaerolineaceae possesses hydrocarbon degradation capabilities and organic acid provision functions, and Terriglobales members exhibit plant growth-promoting potential through direct plant interactions [40,41,42]. This indicates that SP modulates soil carbon and nitrogen cycling by reshaping the soil microbial community, particularly the functional microorganisms associated with carbon, nitrogen, and phosphorus cycling processes.

As the investigation was conducted at four different tea plantations, the within-group variation in bacterial *α*-diversity indices (Shannon, ACE, OBS, and PD indices) may be greater than between-group variation using parametric tests, and this would lead to inappropriate results. Given the limitation of parametric test on this condition, the non-parametric test, through rank-based analyses, was chosen for a more robust statistical inference. The observed α-diversity enhancement under SP aligns with the resource heterogeneity hypothesis [4], wherein organic application created niche opportunities for both oligotrophic and copiotrophic taxa, as demonstrated in tea [16] and tobacco [43] agroecosystems, and can benefit soil quality maintenance and ecosystem functionality [4,16]. Notably, the increase in PD suggests SP significantly enhanced the microbial phylogenetically distinct taxa [44], thus promoting the stronger functional redundancy and environmental adaptability of soil bacterial communities. The *β*-diversity shifts revealed by PCoA (with 36.0% of the variance explained) demonstrate the restructuring of community assembly under SP. The tighter clustering of SP implies the straw inputs reduced within-treatment variability via directional selection and enrichment effects on the bacterial taxa [45].

The LEfSe results revealed the significant shifts in biomarkers under SP, such as the phylum-level enrichment of Nitrospirota. The results indicate the microbial functional restructuring induced by SP, characterized by the selective enrichment of cellulolytic functional guilds critical for straw decomposition [4,46]. The enrichment of Anaerolineae (LDA = 4.11) in SP indicates that anaerobic lignocellulose degraders within Chloroflexota. Conversely, the dominance of Acidobacteriae (LDA = −4.47) in CK reflects their adaptation to oligotrophic conditions caused by the lower carbon availability in non-straw-amended tea cultivation systems.

At the genus level, the enrichment of *MND1* (LDA = 3.82) and the suppression of *Ellin6067* (LDA = −3.63) in S indicate the sensitivity to straw-induced physicochemical changes in the same family, Nitrosomonadaceae, which play a pivotal role in nitrification by converting ammonia to nitrite. Conversely, the enrichment of *Chryseotalea* (LDA = 3.60) and *Pirellula* (LDA = 3.53) in S soils mirrors their documented roles in polysaccharide metabolism in high-carbon environments [47,48]. Notably, the stimulation of *Nitrospira* (LDA = 3.41) correlates with enhanced nitrification potential, consistent with the study on nitrogen-cycling microbiota in tea soils under SP [49].

Straw decomposition releases labile carbon substrates, which broaden niche opportunities for oligotrophic and copiotrophic taxa. The sustained carbon input likely facilitated the proliferation of fast-growing copiotrophs (e.g., Proteobacteria), while maintaining metabolic flexibility for slow-growing specialists [50]. Such resource-mediated diversification is consistent with the microbial priming effect, where organic applications stimulate both taxonomic and functional diversity [51].

The network restructuring under SP revealed a trade-off between functional specialization and generalist dominance (Figure 5). Specifically, the shift from Acidobacteriae_Bryobacterales (CK) to Blastocatellia_Blastocatellales (S) within Acidobacteriota (Appendix A) aligns with the well-known oligotrophic adaptation to low-pH conditions [41], suggesting that SP alleviates acidity stress through organic buffering, thereby enabling Acidobacteriota to transition toward more versatile metabolic strategies. The reduced network complexity aligns with the findings that organic inputs enhance niche partitioning among microbial guilds [52]. This modular architecture likely reflects optimized resource utilization, where specialized functional groups (e.g., decomposers) form cohesive subcommunities to exploit straw-derived carbon. Notably, Pseudomonadota and Acidobacteriota emerged as keystone taxa in S networks, driving among-module interactions. Pseudomonadota, known for lignin and cellulose degradation, likely facilitated straw decomposition [53], while Acidobacteriota, Actinomycetota, and Chloroflexota occupied oligotrophic niches through slow-growth strategies [54,55]. An oligotrophic adaptation strategy for the nutrient and physiochemical changes caused by SP has been suggested. The reduced network complexity and retained high clustering coefficient under SP ensured local connectivity and functional redundancy, with better ecosystem resistance [56]. This trade-off confirmed the microbial adaptation to balance efficiency and stability in straw-amended systems.

The shift with fewer module hubs and more connectors in S, coupled with reduced *Z_i_* and elevated *P_i_*, indicates enhanced within-module connectivity [55,56]. This restructuring promotes metabolic complementarity for organic matter decomposition. Concurrently, the 69.2% increase in negative edges also confirmed the intensified competition for labile carbon, consistent with straw-induced resource heterogeneity [57].

This intensified competition may reflect priority effects among copiotrophs [50], where early colonizers of straw-derived carbon suppressed late-arriving taxa through resource preemption, while cooperative networks (positive cohesion) remained statistically unchanged. The intensified negative cohesion suggests the competitive exclusion of taxa less well adapted to high-resource conditions. SP preferentially enriched copiotrophic lineages (r-strategists) capable of rapid substrate utilization [58], including nitrogen-cycling guilds such as *Nitrospira*, *Pirellula*, *MND1*, and *Rhodocyclaceae*. This aligns with global meta-analyses under organic fertilization [59]. Notably, Zhang et al. [60] specifically documented the straw-induced upregulation of nitrogenase and urease activities in tea soils, driven by the competitive dominance of metabolic generalists. Meanwhile, elevated negative cohesion reflects a self-organized network architecture where competitive exclusion filtered for functionally redundant taxa. Thus, community and functional stability were maintained by metabolic overlap among antagonistic populations under environmental fluctuations [61,62]. As demonstrated in grassland soils by Hassani et al. [63], such networks exhibit superior resistance to pH fluctuations and xenobiotic stress—critical traits for tea soils facing seasonal fertilizer inputs and tea picking. In general, the decoupling between positive/negative cohesion responses (*p* > 0.05 vs. *p* < 0.001) implies that SP restructures microbial interactions through substrate-mediated competition rather than direct facilitation. This mechanistic shift supports the ecological theory that resource availability modulates microbial interaction types along the cooperation–competition continuum [64].

The dbRDA results display the congruent impacts of soil physicochemical properties on *β*-diversity and keystone taxa, which demonstrate that keystone taxa could effectively mirror community assembly patterns. They also reveal that WC and pH were the primary drivers of bacterial *β*-diversity and keystone taxa in straw-amended tea soils. WC dominated community shifts in straw-amended soils, while pH dominated in non-straw-amended soils. The shift in the main driving factor from pH to WC indicates SP alleviates soil pH effects on bacterial communities in tea gardens through the copiotrophic enrichment, oligotrophic specialization, and nitrogen metabolism regulation observed in the ecosystem. In addition, straw mulching reduced soil water evaporation, thereby influencing soil water content and subsequently affecting microbial activity. The results demonstrate the significant impacts of soil water content, likely attributable to the combined effects of alterations in the soil aggregate composition, soil water capacity, and reduced water evaporation induced by straw mulching.

Straw inputs introduced labile and recalcitrant carbon, fostering niche partitioning—copiotrophic Pseudomonadota dominated high-moisture zones, while Acidobacteriota occupied low-pH microsites. This resource heterogeneity enhanced metabolic complementarity, as evidenced by the increased network of among-module connectivity. Optimized SP strategies with region-specific application patterns can synergistically improve microbial network stability and niche differentiation by competition–cooperation trade-off. Thereby, ecosystem multifunctionality is improved through stimulated carbon sequestration and nitrogen retention, which can reinforce abiotic stress resilience, such as drought resistance.

## 5. Conclusions

The study demonstrated the effects of SP on the bacterial communities in the tea plantations of southern Henan. The results demonstrate that straw induced significant shifts in soil bacterial community structure, diversity, and ecological interactions. SP reshaped the dominant phyla (Acidobacteriota, Pseudomonadota, and Chloroflexota) composition, with Nitrospirota and Vicinamibacterales and Anaerolineaceae (*p* < 0.05) significantly increased, and Terriglobales reduced (*p* < 0.05). These taxonomic changes were accompanied by enhanced *α*-diversity (Shannon, ACE, OBS, and PD indices) and restructured *β*-diversity, reflecting a transition toward a more metabolically versatile bacterial consortium. The LEfSe results confirmed the selective enrichment of nitrogen-cycling taxa (Nitrospira), copiotrophs (Chryseotalea), and anaerobic degraders (Anaerolineaceae), alongside oligotrophic lineage suppression (Ellin6067). The co-occurrence networks exhibited intensified competition and simplified complexity, with Pseudomonadota and Acidobacteriota dominating among-module connectivity. Soil WC, pH, and AP explained 58.4% of the *β*-diversity variation, with WC and pH further modulating the keystone taxa assembly. This study highlights that SP enhances the microbial functional redundancy and accelerates nutrient turnover in tea agroecosystems, driven by resource-driven niche partitioning between copiotrophic and oligotrophic taxa, and nitrogen-cycling guilds through the competition–cooperation equilibrium.

## Figures and Tables

**Figure 1 microorganisms-13-00832-f001:**
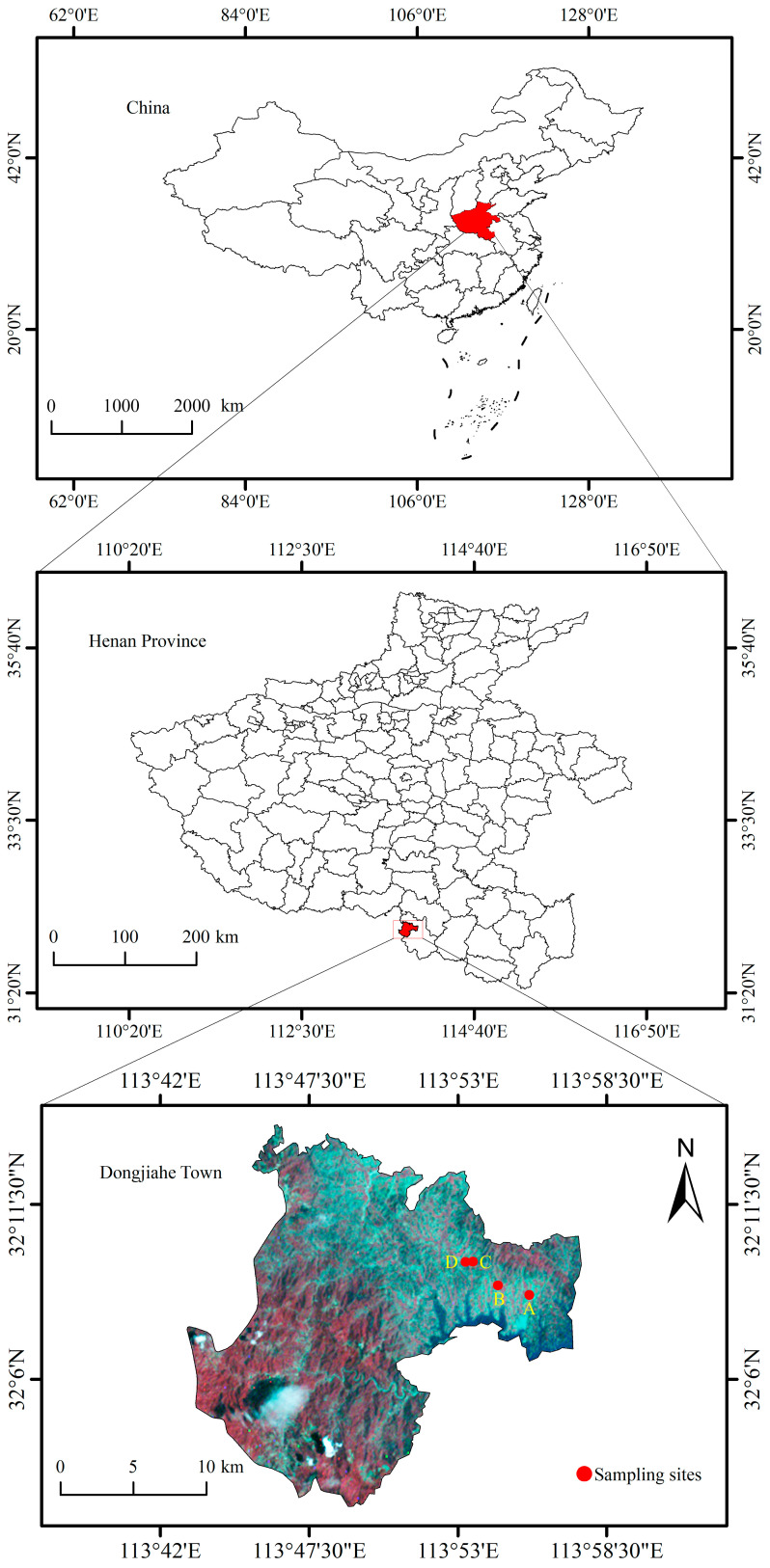
The geographic location of the study area and soil sampling sites. A: tea plantation 1, B: tea plantation 2, C: tea plantation 3, and D: tea plantation 4.

**Figure 2 microorganisms-13-00832-f002:**
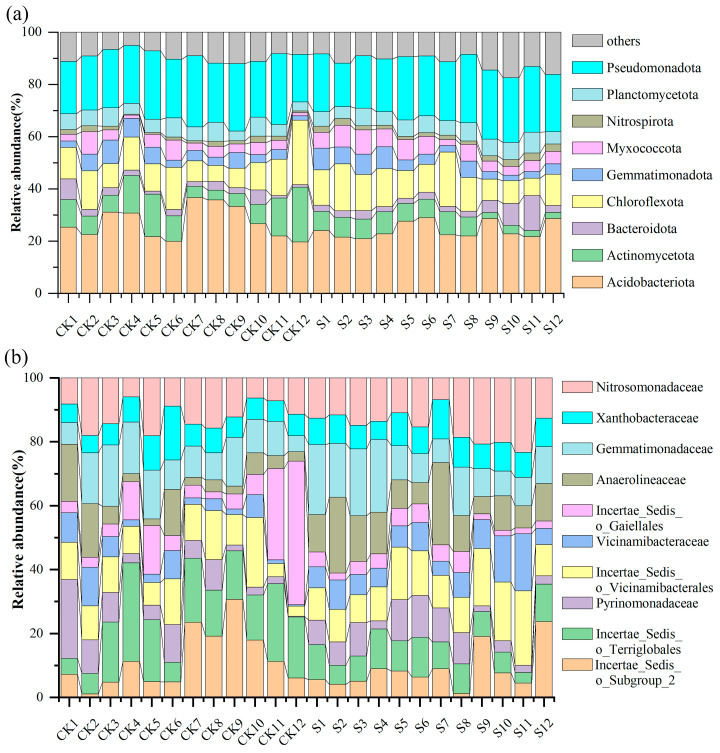
(**a**) The relative abundance of different straw treatments (CK and S) at the phylum level; (**b**) The relative abundance of different straw treatments (CK and S) at the family level. CK, a soil sample from the tea plantations without straw application; S, a soil sample from the tea plantations with straw application.

**Figure 3 microorganisms-13-00832-f003:**
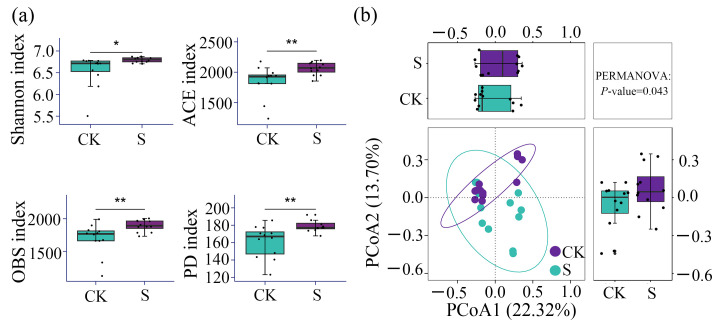
(**a**) The *α*-diversity index comparisons of Shannon, the abundance-based coverage estimator metric (ACE), observed OTUs (OBS), and phylogenetic diversity (PD) between different straw treatments (CK and S); (**b**) The principal coordinate analysis (PCoA) of different straw treatments (CK and S) based on Bray–Curtis distances at the ASV level. CK, a soil sample from the tea plantations without straw application; S, a soil sample from the tea plantations with straw application and permutational multivariate analysis of variance (PERMANOVA). *, significance at the 0.05 level; **, significance at the 0.01 level.

**Figure 4 microorganisms-13-00832-f004:**
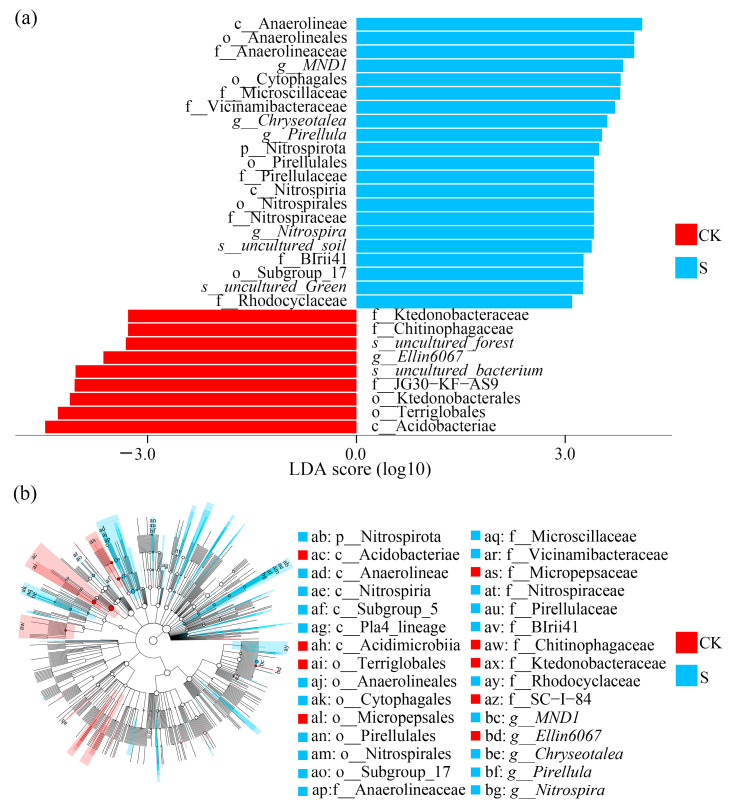
(**a**) The linear discriminant analysis (LDA) results of the top 30 differential bacterial taxa analyzed between CK and S; (**b**) the cladogram results of the top 30 diverse taxa between CK and S generated based on the LDA effect size (LEfse). CK, a soil sample from the tea plantations without straw application; S, a soil sample from the tea plantations with straw application.

**Figure 5 microorganisms-13-00832-f005:**
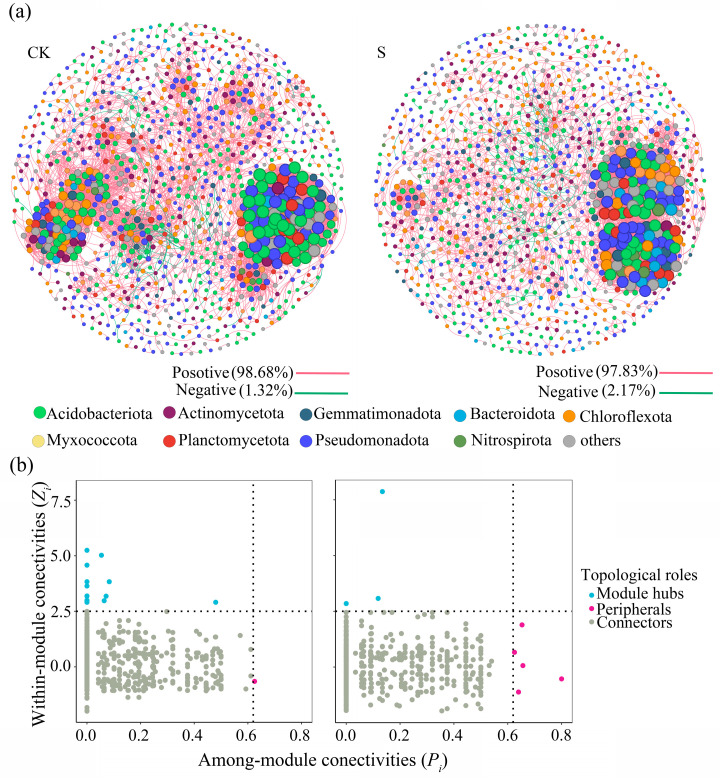
(**a**) The soil bacterial co-occurrence network analysis of CK and S (**a**); (**b**) the within-/among-module connectivity results of the soil bacterial co-occurrence network in CK and S. CK, a soil sample from the tea plantations without straw application; S, a soil sample from the tea plantations with straw application.

**Figure 6 microorganisms-13-00832-f006:**
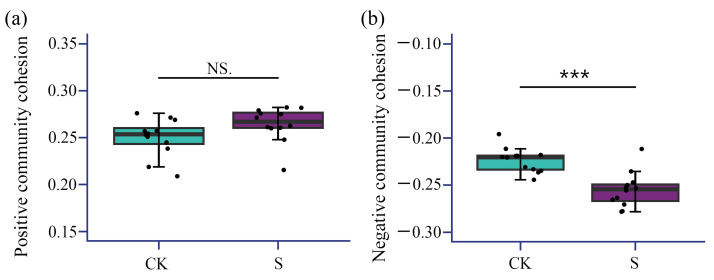
(**a**) The boxplot comparison of positive community cohesion between CK and S; (**b**) the boxplot comparison of negative community cohesion between CK and S. CK, a soil sample from the tea plantations without straw application; S, a soil sample from the tea plantations with straw application. NS., no significance; ***, significance at the 0.001 level.

**Figure 7 microorganisms-13-00832-f007:**
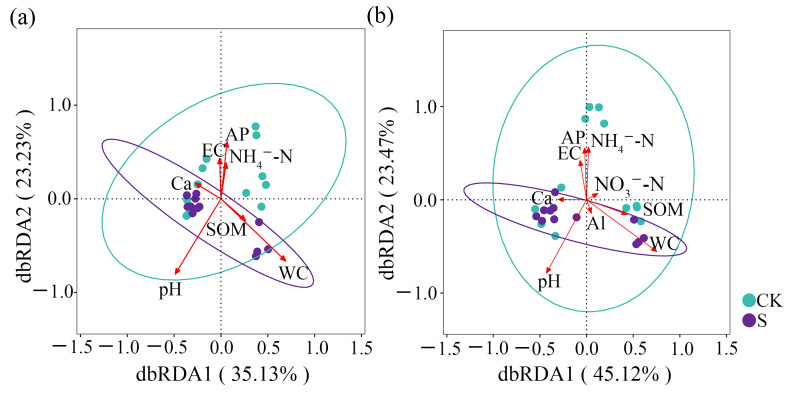
(**a**) The distance-based redundancy analysis (dbRDA) plots of bacterial *β*-diversity with the soil physicochemical properties; (**b**) the dbRDA plots of the keystone taxa with the soil physicochemical properties. CK, a soil sample from the tea plantations without straw application; S, a soil sample from the tea plantations with straw application.

**Table 1 microorganisms-13-00832-t001:** The main topological properties of the co-occurrence network of CK and S.

Parameters	CK	S	S/CK Ratio Variation
node	1048	1020	−2.67%
Edge	9552	6036	−36.8%
Average degree	18.2	11.8	−35.1%
Network diameter	21	19	−9.52%
Graph density	0.017	0.012	−29.4%
Modularity index (*Q*)	0.663	0.687	3.62%
Clustering coefficient (*C*)	0.573	0.530	−7.50%
Positive edges	0.987	0.978	−0.86%
Negative edges	0.013	0.022	69.2%

**Table 2 microorganisms-13-00832-t002:** The phylum proportion of nodes in the co-occurrence network of CK and S.

Phylum	CK Proportion	S Proportion
Acidobacteriota	26.24%	22.25%
Pseudomonadota	25.86%	24.41%
Chloroflexota	11.35%	12.45%
others	10.69%	13.14%
Actinomycetota	9.45%	7.16%
Planctomycetota	6.39%	5.69%
Myxococcota	3.72%	5.78%
Gemmatimonadota	2.96%	4.61%
Bacteroidota	2.77%	3.33%
Nitrospirota	0.57%	1.18%

**Table 3 microorganisms-13-00832-t003:** The hierarchical partitioning (HP) results of the distance-based redundancy analysis (dbRDA) of the keystone taxa of CK and S.

Group	Soil Physicochemical Properties	Unique	Average Share	Individual	Individual Percentage (%)
Bacterial *β*-diversity	WC	0.1125	0.0264	0.1389	24.72
	pH	0.0829	0.0326	0.1155	20.55
	AP	0.0395	0.0249	0.0644	11.46
	OM	0.0366	0.0165	0.0531	9.45
	Ca	0.028	0.0179	0.0459	8.17
	NH_4_^+^-N	0.0303	0.0122	0.0425	7.56
	EC	0.0343	0.004	0.0383	6.81
	Al	0.0249	0.0109	0.0358	6.37
	NO_3_^−^-N	0.0261	0.0019	0.028	4.98
Keystone taxa	WC	0.1396	0.0549	0.1945	30.68
	pH	0.0867	0.0466	0.1333	21.03
	AP	0.0613	0.0142	0.0755	11.91
	NH_4_^+^-N	0.0428	0.0144	0.0572	9.02
	SOM	0.0251	0.0256	0.0507	8
	Ca	0.0178	0.0268	0.0446	7.03
	EC	0.0253	0.0039	0.0292	4.61
	Al	0.0126	0.0149	0.0275	4.34
	NO_3_^−^-N	0.0151	0.0062	0.0213	3.36

## Data Availability

The raw bacterial sequencing data have been deposited in the Genome Sequence Archive (GSA) at the National Genomics Data Center (NGDC), China National Center for Bioinformation/Beijing Institute of Genomics, Chinese Academy of Sciences, under accession number CRA024337; other data are contained within the article.

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
