# Peer review of "Straw-Enhanced Soil Bacterial Robustness via Resource-Driven Niche Dynamics in Tea Plantations, South Henan, China"

_microorganisms, 2025, doi:10.3390/microorganisms13040832_

Round 1

Reviewer 1 Report

Comments and Suggestions for Authors

Words in the title should start from capitalized initials. Please follow the guidelines for authors. Citation numbers are placed as superscript, but should be placed as normal text.

It would be good if the weather condition for year of the study (2018) were presented. In current version of the manuscript only average climatic conditions are presented.

Please provide more details about straw mulching. What type of straw was used and 15,000 kg per ha it is in dry matter or fresh matter?

Please provide basic information about soil type (e.g. based on FAO classification) and soil texture class (or sand, silt, clay content). It is important because soil type affects strongly microbial community.

Table 3. Some variables are not soil properties, e.g. water content vary depending on current conditions. After rain water content is very high and after longer dry period water content in soil is much lower but it not affects soil properties. Soil water capacity can be important soil property but not water content.

In material and methods lack information how these soil properties were determined and in what units are these soil properties. Please be more specific in presentation of the results.

In the Data Availability Statement there is information that “Data are contained within the article”. In the manuscript there are results not the raw data. Please notice that journal policy requires the free availability of the raw data.

Author Response

Comments 1: Words in the title should start from capitalized initials. Please follow the guidelines for authors.

Response 1: Thank you for pointing this out. We had capitalized the initials of the words in the title. See Page 1 line 2-3.

Comments 2: Citation numbers are placed as superscript, but should be placed as normal text.

Response 2: Thank you for pointing this out. We had placed the citation numbers as normal text.

Comments 3: It would be good if the weather condition for year of the study (2018) were presented. In current version of the manuscript only average climatic conditions are presented.

Response 3: Agree. The meteorological data (e.g., temperature, precipitation and humidity) presented in the manuscript represent the 2017–2020 averages. We have presented the weather condition for year of the study (2018) in the manuscript, and supplemented the specific meteorological data for 2017-2018 in Table S1. See page 2 line 87-89.

Comments 4: Please provide more details about straw mulching. What type of straw was used and 15,000 kg per ha it is in dry matter or fresh matter?

Response 4: Done. We had provided the details about straw mulching, such as straw types, length in the different tea plantation in the manuscript, with the straw characteristics provided in Table S2. See page 3 line 101-103, and Figure 1. And, 15,000 kg per ha is the fresh matter of the straw, which was displayed in line 105 of the manuscript.

Comments 5: Please provide basic information about soil type (e.g. based on FAO classification) and soil texture class (or sand, silt, clay content). It is important because soil type affects strongly microbial community.

Response 5: Done. We have added the soil type and texture class in the line 91-92 of the manuscript.

Comments 6: Table 3. Some variables are not soil properties, e.g. water content vary depending on current conditions. After rain water content is very high and after longer dry period water content in soil is much lower but it not affects soil properties. Soil water capacity can be important soil property but not water content.

Response 6: Thank you for pointing this out. The selection of soil sampling in late April was based on the relatively stable precipitation during this period, which is comparable to over half of the months annually. The sampling time ensures that soil water content reflects soil water capacity to a certain extent, as evidenced by relevant literature demonstrating a significant correlation between soil water content and water-holding capacity (Liu, S.X.; Mao, L.X.; Mo, X.G.; Zhao, W.M.; Lin, Z.H. Analysis of spatial variability of soil moisture and its driving force factors in the Shaanxi-Henan region along the Yellow River. Clim. Environ. Res. 2008, 13(5), 645–657. (In Chinese).). Additionally, straw mulching reduced soil water evaporation, thereby influencing soil water content and subsequently affecting microbial activity. Consequently, this study adopted soil water content as a key indicator. The results demonstrated significant impacts of soil water content, likely attributable to combined effects of alterations in soil aggregates composition, soil water capacity and reduced water evaporation induced by straw mulching. We also add the analysis in the “2.1 Description of Sites and Sampling” and the “4. Discussion” of the manuscript. See page 3 line 98-101, and page 13-14 line 406-410

Comments 7: In material and methods lack information how these soil properties were determined and in what units are these soil properties. Please be more specific in presentation of the results.

Response 7: Done. We have added the references of determination of soil properties, such as WC, AP in the “Material and Methods”. See page 5 line 121 and 127. The soil properties of the different samples were provided in Table S3.

Comments 8: In the Data Availability Statement there is information that “Data are contained within the article”. In the manuscript there are results not the raw data. Please notice that journal policy requires the free availability of the raw data.

Response 8: Done. We have added the data about “The characteristics of the straw”, “the soil properties of different samples” and “The community composition of dominant bacteria” in Table S2, Table S3 and Table S4.

Reviewer 2 Report

Comments and Suggestions for Authors

The manuscript microorganisms-3531350 describes a study of the effect of straw on the soil bacterial community on a tea plantation. The authors have done significant bioinformatics work to analyze the results, but I find that the manuscript has several weaknesses that most likely cannot be corrected, but they should be discussed by the authors in the text of the manuscript.

Straw is an organic substance of complex composition. The authors do not provide any characteristics of the straw used: neither the plant from which it was obtained; nor the methods of obtaining; nor the chemical characteristics. The lack of characteristics of the straw (some straw) makes it impossible to reproduce the described experiment, i.e. significantly reduces the scientific value of such a publication.

The authors claim that straw produces a shift in the microbial community. But straw contains its own microbiota, which also gets into the analyzed soil and can lead to the changes in microbial diversity observed by the authors. The minor changes in the community structure noted by the authors may be due to the introduced microbes, and not to the structuring role of the straw itself.

Tea is a perennial culture, so adding 15 t/ha of straw mulching without damaging the root system of plants is not a trivial task. The authors do not describe the technology of adding straw to soil with perennial cultures in any way.

Minor remarks:

Lines 42-46: The information here is very dense. Explain in more detail.

Line 63: "one of China's Top Ten Famous Teas" - provide a reference(s) to this information.

Line 79-90: Provide references to the information provided.

Subsection 3.2: Specify the relative change for all compared indicators (in percent).

Figure 3a: In each of the graphs, 1–2 points for CK are very different from the mean and other points. Did the authors test whether these points belong to the sample (normality or χ2 test)?

Figure 3b: Provide values for the axes of the graphs.

Figure 3 and 6 captions: Provide the meaning of the asterisks.

Line 182: The authors draw conclusions about the entire variation based on the first two principal coordinates, which explain only 36% of the variation. How relevant is this conclusion?

Line 193: Figure 4a provides labels for LDA score -2.5 and 2.5. For clarity, change the labels to LDA>3.0 (as in the text).

Lines 196-200: The LDA scores are almost identical for class, order, and family. Does this indicate a contribution from members of the same family? If so, why include all 3 taxonomic ranks?

Lines 232-234: Estimate the statistical significance of the changes.

Tables 1 and 2: Add an estimate of the statistical significance of the differences.

Figure 3 and Table 3: Use abbreviations in figure and table captions.

Lines 282-284: Is straw a source of nitrites and ammonia? And how does oxidation of ammonia to nitrites improve nitrogen supply to plants?

Lines 285-286: Indicate how much abundances increased and decreased.

Line 345: Instead of reference [4], provide a reference to lignin and cellulose degraders here.

Line 382: The authors discuss the great importance of pH and WC in the manuscript, but do not provide absolute values for these parameters during the experiment.

Lines 387-388: What experimental data (Figures or Tables) are presented in the manuscript that support this assertion? And what do the authors mean by "dominated"? In Figure 2, Pseudomonadota does not dominate in any sample.

Author Response

Comments 1: The manuscript microorganisms-3531350 describes a study of the effect of straw on the soil bacterial community on a tea plantation. The authors have done significant bioinformatics work to analyze the results, but I find that the manuscript has several weaknesses that most likely cannot be corrected, but they should be discussed by the authors in the text of the manuscript.

Response 1: Thank you. We have added some information and rewritten some paragraphs in “Introduction”, “Methods and Materials”, “Results” and “Discussion” of the manuscript to solve the weaknesses.

Comments 2: Straw is an organic substance of complex composition. The authors do not provide any characteristics of the straw used: neither the plant from which it was obtained; nor the methods of obtaining; nor the chemical characteristics. The lack of characteristics of the straw (some straw) makes it impossible to reproduce the described experiment, i.e. significantly reduces the scientific value of such a publication.

Response 2: Done. We have added the information about the straw applied in the tea plantations,”Wheat straw mulching was applied between the tea plant rows in the tea plantations 1 and 3, whereas rice straw mulching was used in the tea plantations 2 and 4 (Table S2). Both the wheat and rice straw length were controlled with 15–20 cm.”. See page 3 line 101-103 and line 105. The tea plantations also were remarked in Figure1. Meanwhile, the physical and chemical characteristics of the straw were also supplemented in Table S2.

Comments 3: The authors claim that straw produces a shift in the microbial community. But straw contains its own microbiota, which also gets into the analyzed soil and can lead to the changes in microbial diversity observed by the authors. The minor changes in the community structure noted by the authors may be due to the introduced microbes, and not to the structuring role of the straw itself.

Response 3: Thank you for pointing this out. We have added the description “Although straw application introduces exogenous microorganisms, studies have shown its primary influence on soil microbial communities is mediated through alterations in soil physicochemical properties rather than the direct effects of these external microbes, for the resilience of the soil microbials” in “1. Introduction”, “… and five subsamples (0–20 cm depth)…”had been modified to “After removing the 0-5 cm topsoil to eliminate interference from external environmental factors, five subsamples (5–20 cm depth) per plot…” in “2. Methods and Materials”, and “The soil microbiome demonstrates strong resilience, with straw-induced shifts primarily driven by straw characteristics rather than exogenous microbial inputs, as the latter fail to persist under prolonged soil conditions.” in “4. Disscussion”. See page 1 line 38-42, page 3 line 107-108 and page 11 line 302-305.

Comments 4: Tea is a perennial culture, so adding 15 t/ha of straw mulching without damaging the root system of plants is not a trivial task. The authors do not describe the technology of adding straw to soil with perennial cultures in any way.

Response 4: Done. See Response 2.

Minor remarks:

(1)Lines 42-46: The information here is very dense. Explain in more detail.

Response: Agree. We have revised the description. “long-term monocropping in tea plantations results in reduced soil disturbance, while acidic soil conditions impose strong selective pressures on soil microbiota [6, 7], and caused the tea soils characterized…” See page 2 line 48-50.

(2)Line 63: "one of China's Top Ten Famous Teas" - provide a reference(s) to this information.

Response: Done. We have added the reference “Wang, H.J.; Cao, X.L.; Yuan, Z.F.; Guo, G.Y. Untargeted metabolomics coupled with chemometrics approach for Xinyang Maojian green tea with cultivar, elevation and processing variations. Food Chem. 2021, 352, 129359. https://doi.org/10.1016/j.foodchem.2021.129359.”. See page 2 line 69.

(3)Line 79-90: Provide references to the information provided.

Response: Done. We have added two references about the information provided in line 79-90. In accordance with Reviewer 1's recommendations, we have revised the meteorological data for the study year 2018 instead of the average data of 2017-2020, which sourced from the Xinyang Meteorological Bureau. We also add the reference about the study area information. See page 2 line 87-89, page 3 line 92 and Table S1. The fertilization standards are based on local practical conditions.

(4)Subsection 3.2: Specify the relative change for all compared indicators (in percent).

Response: Agree. We have added. “S increased the Shannon, ACE, OBS, and PD indices by 3.7%, 12.0%, 12.1%, and 11.2%, respectively.” See page 7 line 192-193.

(5)Figure 3a: In each of the graphs, 1–2 points for CK are very different from the mean and other points. Did the authors test whether these points belong to the sample (normality or χ2 test)?

Response: Thank you for pointing this out. The sample distribution was non-normally distributed; nonparametric tests were utilized, and related descriptions and figures have been updated. According to this, “… t-test,” had been modified to “… Wilcox -test,”, “… Shannon index (P < 0.05), ACE estimator (P < 0.05), and OBS richness (P < 0.05). Notably, phylogenetic diversity (PD) exhibited the most pronounced enhancement under S treatment (P < 0.01).” had been modified to “… Shannon index (P < 0.05). Notably, ACE estimator, OBS richness and phylogenetic diversity (PD) exhibited the most pronounced enhancement under S treatment (P < 0.01).”. See page 5 line 157, and page 7 line 194-196, and Figure 3a.

(6)Figure 3b: Provide values for the axes of the graphs.

Response: Done. The axes values of the box plot in Figure 3b were the same with the axes of Principal coordinate analysis (PCoA) in Figure 3b. We have added the values of the axes of the graphs, and updated Figure 3b.

(7)Figure 3 and 6 captions: Provide the meaning of the asterisks.

Response: Done. The meaning of the asterisks was provided in the figure cations. See page 7 line 209-210 and page 10 line 277.

(8)Line 182: The authors draw conclusions about the entire variation based on the first two principal coordinates, which explain only 36% of the variation. How relevant is this conclusion?

Response: Thank you. High-throughput sequencing of microbial communities yielded 100,690 high-quality sequences, clustered into 22,668 amplicon sequence variants (ASVs). PCoA illustrated sample distribution in reduced-dimensional space, with principal coordinates 1 and 2 explaining the majority of the community variation, respectively. This analytical framework is widely validated in microbial ecology studies, such as “Wang, Z.H.; Li, Z.W.; Zhang, Y.J.; Liao, J.Y.; Guan, K.X.; Zhai, J.X.; Meng, P.F.; Tang, X.L.; Dong, T.; Song, Y. Root hair developmental regulators orchestrate drought triggered microbiome changes and the interaction with beneficial Rhizobiaceae. Nat. Commun. 2024, 15, 54417. https://doi.org/10.1038/s41467-024-54417-5.”. Given the high dimensionality of microbial community data (over 20,000 features), reducing it to two dimensions retains a good representativeness with 36% cumulative variance explained.

(9)Line 193: Figure 4a provides labels for LDA score -2.5 and 2.5. For clarity, change the labels to LDA>3.0 (as in the text).

Response: Done. We have been changed the labels of Figure4a to LDA > 3.0 as in the text, and updated Figure 4a.

(10)Lines 196-200: The LDA scores are almost identical for class, order, and family. Does this indicate a contribution from members of the same family? If so, why include all 3 taxonomic ranks?

Response: Thank you. The LDA scores are identical in different taxa levels, e.g. Nitrospirota at the phylum level, Anaerolineae at the class level, Anaerolineales at the order level, Ktedonobacteraceae at the family level, and Chryseotalea at the genus level. Due to the extensive dataset, we emphasize the genus-level analysis, such as “MND1” and ”Ellin6067” which both belong to the family of “Nitrosomonadaceae”, while “Acidothermus” belong to the family of “Acidothermaceae”, which was displayed at the Table S5.

(11)Lines 232-234: Estimate the statistical significance of the changes.

Response: Thank you. Co-occurrence network analysis primarily aims to identify association patterns (e.g., synergy or competition) among microbial community taxa, serving as an exploratory tool rather than hypothesis testing. It focuses on characterizing global network topological features (e.g., modularity, hub nodes) rather than assessing the statistical significance of individual associations (Kajihara, K.T.; Hynson, N.A. Networks as tools for defining emergent properties of microbiomes and their stability. Microbiome  2024, 12, 1868. https://doi.org/10.1186/s40168-024-01868-z.). Meanwhile, given the high-dimensional nature of microbial data (e.g., over 20,000 ASVs in this study), conventional significance testing methods (e.g., p-value correction) face severe multiple hypothesis testing challenges in sparse matrices, leading to elevated false positive rates. Instead, the SPIEC-EASI algorithm was used to generate the network data, which inherently incorporates significance assessment for microbial associations. Consequently, no additional conventional significance testing was applied to the network data.

(12)Tables 1 and 2: Add an estimate of the statistical significance of the differences.

Response: Thank you. The data in Table 1 and Table 2 were obtained based on the network generated by SPEIC EASI, and no additional conventional significance testing was applied, according to Response of Minor remark (11).

(13)Figure 3 and Table 3: Use abbreviations in figure and table captions.

Response: Agree. We have added “PCoA Principal coordinate analysis” and “dbRDA Distance-based redundancy analysis.” in “Abbreviations”.

(14)Lines 282-284: Is straw a source of nitrites and ammonia? And how does oxidation of ammonia to nitrites improve nitrogen supply to plants?

Response: Agree. We have added the description about how the oxidation of ammonia to nitrites to improve nitrogen supply to plants. “…by reduced the ammonia volatilization and increase the nitrate nitrogen…”. See page 12 line 310-311.

(15)Lines 285-286: Indicate how much abundances increased and decreased.

Response: Done. We have revised the description. “At the family level, S-treated soil showed significant increases in Vicinamibacteraceae and Anaerolineaceae (P < 0.05), while o_Terriglobales_Incertae_edis exhibited decreased abundance (P < 0.05).” had been modified to “ At the family level, S-treated soil showed significant increases in Vicinamibacteraceae and Anaerolineaceae (P < 0.05) by 63.7% and 81.4%, while o_Terriglobales_Incertae_edis exhibited decreased abundance (P < 0.05) of 50.2%.”. See page 13 line 312-314.

(16)Line 345: Instead of reference [4], provide a reference to lignin and cellulose degraders here.

Response: Done. We have been added a reference to lignin and cellulose degraders. “Qi, L.L.; Yuan, J.; Zhang, W.J.; Liu, H.Y.; Li, Z.P.; Bol, R.; Zhang, S.X. Metagenomics reveals the underestimated role of bacteria in the decomposition of downed logs in forest ecosystems. Soil Biol. Biochem. 2023, 187, 109185. https://doi.org/10.1016/j.soilbio.2023.109185.”

(17) Line 382: The authors discuss the great importance of pH and WC in the manuscript, but do not provide absolute values for these parameters during the experiment.

Response: Thank you for pointing this out. We have added the soil properties in Table S3.

(18)Lines 387-388: What experimental data (Figures or Tables) are presented in the manuscript that support this assertion? And what do the authors mean by "dominated"? In Figure 2, Pseudomonadota does not dominate in any sample.

Response: Thank you. Pseudomonadota was one of the dominant bacterial phyla based on the data of community compositions and co-occurrence network. And the modulehubs and connectors results in Table S6 also revealed the keystone taxa shifted from Acidobacteriota and Chloroflexota of CK (33.3% and 22.2%, respectively), to Pseudomonadota and Acidobacteriota of S (44.4% and 33.3%, respectively). The results were displayed also in “3.4 Soil Bacterial Co-occurrence Network Analysis” of the manuscript. See page 9 line 248-252.

Other changes in the revised manuscript:

(1)The paragraphs “LEfSe analysis demonstrated that SP induced significant restructuring of …conventional tea monoculture systems[4]” in “4. Disscussion”, which was placed in line 310-317 of the former manuscript was deleted for the repetitive description about the discussion of LEfSe results. See page 12 line 330-331.

(2) The “Zi” and “Pi” in Figure 4b was changed to “Zi” and ”Pi”, and the figure was updated.

Reviewer 3 Report

Comments and Suggestions for Authors

Present manuscript looks good. The use of SP as soil regenaration and soil improve microbial microbial community is very interesting. The correlation among specific microbial population and many physical chemistry such as Soil WC, pH, and AP is clear and high contribute fo our understanding of soil quality and microbial populations. The whole manuscript is ok, I have not any suggestion for the Authors.

Author Response

Comments 1: Present manuscript looks good. The use of SP as soil regenaration and soil improve microbial microbial community is very interesting. The correlation among specific microbial population and many physical chemistry such as Soil WC, pH, and AP is clear and high contribute fo our understanding of soil quality and microbial populations. The whole manuscript is ok, I have not any suggestion for the Authors.

Response 1:Thank you very much. The manuscript had been revised.  

Round 2

Reviewer 1 Report

Comments and Suggestions for Authors

All my comments were addressed. However, I still have one doubt connected with data availability. In the supplementary materials there are the results based on the data but there are no raw data. Will you make the raw data available?

Author Response

Comments 1: All my comments were addressed. However, I still have one doubt connected with data availability. In the supplementary materials there are the results based on the data but there are no raw data. Will you make the raw data available?

Response 1: Agreed. We have deposited the raw bacterial sequencing data in the Genome Sequence Archive (GSA) at the National Genomics Data Center (NGDC), China National Center for Bioinformation / Beijing Institute of Genomics, Chinese Academy of Sciences, under accession number CRA024337. See https://bigd.big.ac.cn/gsa/browse/CRA024337.

Reviewer 2 Report

Comments and Suggestions for Authors

The authors have made all the necessary corrections in the revised manuscript. However, I still think that some of the authors' statements in the manuscript require confirmation in a larger experiment for better statistical support. For example, the conclusion about the effect of straw treatment on alpha diversity is based on the very low value of 1-2 points in the control treatments (CK, Fig. 3a). The remaining points of the control (CK) are no different from the points with the straw treatment (S). And in general, the ranges of the experimental and control parameter values ​​overlap very much, even in cases where the authors note the statistical significance of the differences between the treatments. I find that such weaknesses of the manuscript should be mentioned by the authors in the Discussion and/or Conclusion sections.

Minor remarks:
Table S4: 1) What does "20.8∼6.6" mean for Chloroflexota, S(%)? Why isn't "6.6∼20.8"?
2) What is the principle of ordering families in Top 10 families? It would be more convenient if families were ordered based on CK (%) values.

Author Response

Comments 1: The authors have made all the necessary corrections in the revised manuscript. However, I still think that some of the authors' statements in the manuscript require confirmation in a larger experiment for better statistical support. For example, the conclusion about the effect of straw treatment on alpha diversity is based on the very low value of 1-2 points in the control treatments (CK, Fig. 3a). The remaining points of the control (CK) are no different from the points with the straw treatment (S). And in general, the ranges of the experimental and control parameter values overlap very much, even in cases where the authors note the statistical significance of the differences between the treatments. I find that such weaknesses of the manuscript should be mentioned by the authors in the Discussion and/or Conclusion sections.

Response 1: Thank you. As the investigation was conducted in 4 different tea plantations, the within-group variation of bacterial α-diversity maybe greater than between-group variation, if used parametric tests, which may lead inappropriate results. And Shapiro-Wilks test was also conducted to confirm the normality distribution of bacterial α-diversity, and all the indices of α-diversity was non-normality. Based on these, non-parametric test, through rank-based analyses, was chosen for the more robust statistical inference. We also add some description about the α-diversity in “2.4 Data Analysis” and “4 Discussion”. See page 5 line 162-163 and page 13 line 332-336.

Minor remarks:

1: What does "20.8∼6.6" mean for Chloroflexota, S (%)? Why isn't "6.6∼20.8"?

Response: Thank you for pointing this out. We have revised the data in Table S4.

2: What is the principle of ordering families in Top 10 families? It would be more convenient if families were ordered based on CK (%) values.

Response: Agree. It would be more convenient if families were ordered based on CK (%) values. However, choosing the top 10 families only by CK (%) values may also lead the effects of straw application masked for the low abundance families in CK, which was enriched by straw application. Based on these, top 10 families were chosen based on the total absolute abundance in all samples.